# Omnidirectional propulsion in a metachronal swimmer

**Adrian Herrera-Amaya** [ID] [●], **Margaret L. Byron** [ID] *[●]

Department of Mechanical Engineering, Penn State University, University Park, Pennsylvania, United States of America

[●] These authors contributed equally to this work.
* mzb5025@psu.edu

**Data Availability Statement:** Raw and processed data, as well as analysis code, are available via the Penn State Data Commons at https://doi.org/10.26208/C2ZD-SZ22.

**Funding:** This work was supported by the National Academies Keck Futures Initiative (#DBS5 to MLB);

## Abstract

Aquatic organisms often employ maneuverable and agile swimming behavior to escape from predators, find prey, or navigate through complex environments. Many of these organisms use metachronally coordinated appendages to execute complex maneuvers. However, though metachrony is used across body sizes ranging from microns to tens of centimeters, it is understudied compared to the swimming of fish, cetaceans, and other groups. In particular, metachronal coordination and control of multiple appendages for three-dimensional maneuvering is not fully understood. To explore the maneuvering capabilities of metachronal swimming, we combine 3D high-speed videography of freely swimming ctenophores (*Bolinopsis vitrea*) with reduced-order mathematical modeling. Experimental results show that ctenophores can quickly reorient, and perform tight turns while maintaining forward swimming speeds close to 70% of their observed maximum—performance comparable to or exceeding that of many vertebrates with more complex locomotor systems. We use a reduced-order model to investigate turning performance across a range of beat frequencies and appendage control strategies, and reveal that ctenophores are capable of near-omnidirectional turning. Based on both recorded and modeled swimming trajectories, we conclude that the ctenophore body plan enables a high degree of maneuverability and agility, and may be a useful starting point for future bioinspired aquatic vehicles.

## Author summary

Metachronal swimming—the sequential, coordinated beating of appendages arranged in a row—exists across a wide range of sizes, from unicellular organisms (micrometers) to marine crustaceans (tens of centimeters). While metachronal swimming is known to be scalable and efficient, the level of maneuverability and agility afforded by this strategy is not well understood. This study explores the remarkable 3D maneuverability of ctenophores (comb jellies), and the appendage control strategies they use to achieve it. Ctenophores have eight rows of appendages (instead of the one or two found in crustaceans and other organisms). This higher number of appendages, their distribution along the body, and the independent frequency control between paired rows enables near-

the National Science Foundation (NSF CBET-2120689 to MLB); and the Consejo Nacional de Ciencia y Tecnologia (CONACYT) of Mexico (#2019-000021-01EXTF-00633 to AHA). MLB received salary from NSF. CONACYT and NSF both provided a partial graduate assistantship for AHA. The funders had no role in study design, data collection and analysis, decision to publish, or preparation of the manuscript.

**Competing interests:** The authors have declared that no competing interests exist.

omnidirectional swimming and turning performance, placing ctenophores among the most maneuverable swimmers. We use experiments and mathematical modeling to explore both the real and theoretical performance landscape of the ctenophore body plan, and show that ctenophores are capable of executing tight turns at high speeds in nearly any plane. This omnidirectional swimming capability gives insight into the ecology and behavior of an important taxonomic group, and shows the potential of metachronal swimming as a source of design inspiration for robotic vehicles (particularly those that must navigate complex environments).

## Introduction

Metachronal coordination of appendages is seen in many aquatic organisms spanning a wide range of sizes and body plans, including shrimp, krill, polychaetes, and even aquatic insects [1,2]. Organisms sequentially actuate a row of appendages (pleopods, ctenes, legs, parapodia, cilia, *et al*) to generate fluid flow via drag-based paddling; hydrodynamic interactions between the paddles can improve overall efficiency and flow speed [3]. The generated flow can be used for swimming or for pumping to aid in feeding, clearance of wastes, and other functions [4,5]. This technique is highly scalable, with metachronally coordinated appendages ranging from microns to centimeters in length. Studies of metachronal locomotion have thus far focused primarily on overall swimming ability [6–9], but some metachronal swimmers are also capable of surprising agility. Here, we examine a highly maneuverable and agile metachronal swimmer: ctenophores, or comb jellies.

Ctenophores swim at Reynolds numbers on the order of 1–1000 [6]; both inertia and viscosity impact their movements considerably. Locomotion is driven via eight antiplectic metachronally coordinated rows of paddles (ctenes), which are made up of bundled millimeter-scale cilia [10]. Fig 1 shows the eight ctene rows circumscribing a lobate ctenophore's approximately spheroidal body and its general morphology. The coordination between ctene rows allows ctenophores to turn tightly around many axes, but not their axis of symmetry—that is, ctenophores can yaw and pitch, but they cannot roll. This is not the case for all swimmers: animals that rely on paired appendages or a single row of appendages tend to display maximum turning performance around a single axis, depending on the appendages' positions along the body [11–13]. Some swimmers exploit the flexibility of their bodies to turn, but these usually have anisotropic bending characteristics, and thus have a preferential turning direction [14,15]. Only a small number of animals have completely axisymmetric bending characteristics, which allow them to turn across the entire range of pitching and yawing motions; jellyfish are one example [16], and even jellyfish cannot effectively rotate about their axis of symmetry. However, the single-jet propulsion used by jellyfish medusae has a notable disadvantage: with this strategy, an animal cannot easily reverse its swimming direction. Ctenophores, by contrast, can quickly reverse their swimming direction by changing the power stroke direction of their ctenes [17].

Though ctenophores are primarily planktonic, they also swim actively and are capable of agile maneuvering as described above. However, their turning behavior has only been described qualitatively [18]. Existing quantitative information on ctenophore swimming trajectories comes from single-camera (2D) experiments, and has focused on straight swimming [6,19–21]. Using a single camera limits the analysis to a single plane; if an animal is turning out of the plane of the camera's field-of-view, only the projection of the turn into 2D may be observed. Ctenophores and many other swimmers perform highly three-dimensional

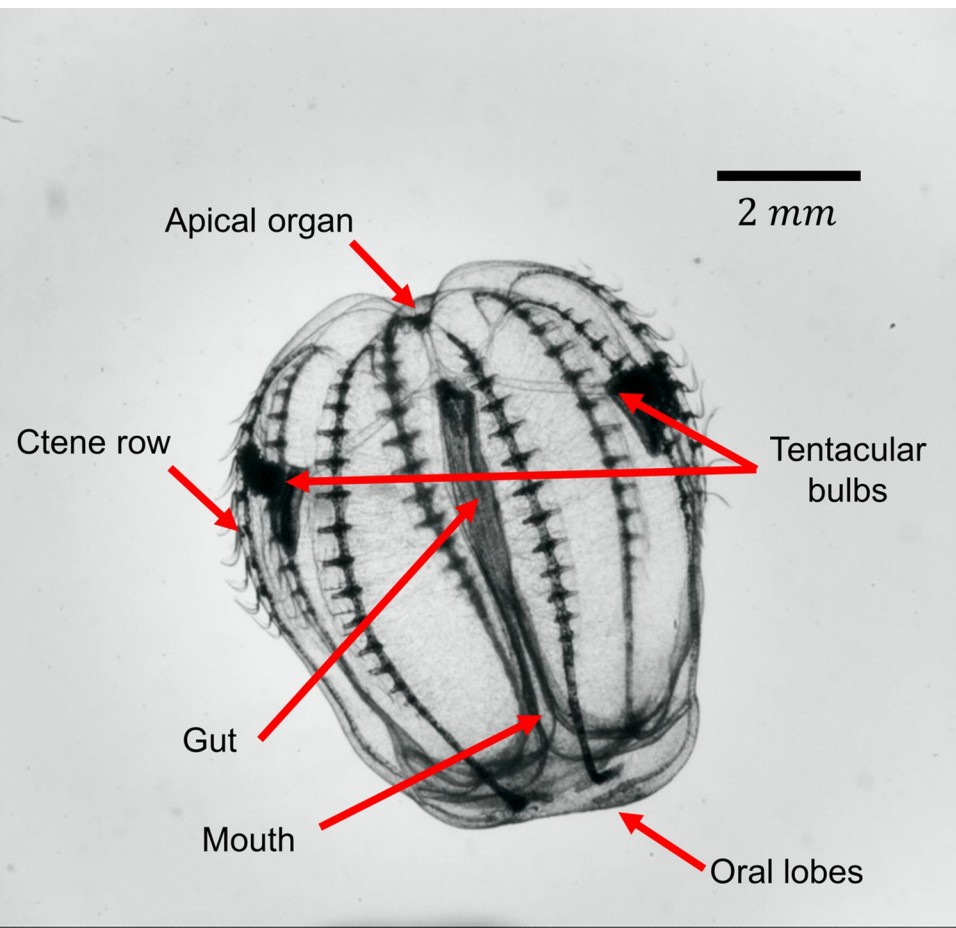

**Fig 1. Morphology of the ctenophore *Bolinopsis vitrea*.**

maneuvers, so a two-dimensional analysis yields only incomplete information (and may dramatically underestimate agility and maneuverability). There is no explicit quantitative data on ctenophores' turning, nor any direct measurements of ctene beating frequencies in the context of turning. Full three-dimensional turning information is not available for any metachronal swimmers, though several organisms have been noted for their agility [9]. We therefore know little of the control strategies—that is, the coordination of frequencies and other parameters between rows—used by ctenophores (or other metachronal swimmers) while performing turning maneuvers.

Two important variables describe turning performance: maneuverability and agility. Maneuverability refers to the ability to turn sharply within a short distance and is typically quantified by the swimming trajectory's radius of curvature (usually normalized by body length) [22]. Agility, however, is not clearly or consistently defined in the animal locomotion literature. A widely used definition is the ability to rapidly reorient the body [23], quantified by the maximum observed angular velocity. However, the angular velocity on its own does not speak to whether the animal needs to stop or slow to perform a turn, which is another colloquial definition of agility. An animal's translational speed while performing a turn can give insight into its agility [13,24,25]. Here, we will use the average speed during the turn ($\overline{V}$) as a measure of agility, and the average normalized radius of curvature ($\overline{R/L}$, where $R$ is the radius

of curvature and $L$ is the body length) during the turn as a measure of maneuverability. We can examine a large number of discrete turns to build a Maneuverability-Agility Plot (MAP), plotting $\overline{R/L}$ *vs.* $\overline{V}$ for a given organism.

In this study, we explore the three-dimensional maneuverability and agility of freely swimming ctenophores, and the control strategies used to produce the observed trajectories. We use multicamera high-speed videography and three-dimensional kinematic tracking to correlate overall trajectories with the beating frequencies of the ctene rows, and identify three distinct turning modes. We also use a 3D reduced-order analytical model to explore the kinematics resulting from the range of physically possible beat frequencies for each turning mode. We use the MAP to explore the observed and hypothetical turning performance of ctenophores, showing how they can sharply turn at high speeds relative to their top speed. In addition, by reconstructing *Bolinopsis vitrea*'s "reachable space," also known as the Motor Volume (MV) [26], we show that ctenophores have the potential to reorient in almost any direction within a small space over a short timeframe (omnidirectionality). Our experimental and analytical results also provide a basis for comparison to other animals known to have high agility and maneuverability, and suggest that ctenophores are worthy of further study as a model for the development of small-scale bioinspired underwater vehicles.

## Materials and methods

### Animal collection and husbandry

*Bolinopsis vitrea* were individually collected in glass jars at Flatt's Inlet, Bermuda, in May 2018 and transported to the Bermuda Institute of Ocean Sciences. The glass collection jars were partially submerged in an open sea table, with filtered seawater flowing continuously around the jars to provide consistent temperature. Experiments were conducted within 12 hours of animal collection, at ambient temperature (21–23˚C). The data presented here are contemporary with those presented in [27].

### Kinematic tracking

Freely swimming animals were filmed synchronously with three orthogonal high-speed cameras (Edgertronic, Sanstreak Corp., San Jose, CA, USA), providing three-dimensional swimming trajectories. The cameras observed an experimental volume of 30×30×30 $mm^3$ (Fig 2A), with each at a framerate of 600 Hz and a resolution of 1024×912 pixels. Each camera was equipped with a 200 $mm$ macro lens (Nikon Micro-Nikkor, Nikon, Melville, NY, USA), with apertures set to f/32 (depth of field ~12 $mm$). Two collimated LED light sources were used to illuminate the volume (Dolan-Jenner Industries, Lawrence, MA, USA). The camera views were calibrated by translating a wand with a micromanipulator through 27 pre-mapped points, creating a virtual 3×3×3 cube. Using the deep learning features of DLTdv8 [28], we pursued a markerless tracking approach using the apical organ and the two tentacular bulbs of the ctenophores (Fig 2B). We recorded 27 free-swimming sequences from eight individuals (*B. vitrea*). We note that the camera system is also described in [29] and [30].

### Ctenophore morphometric and kinematic parameters

To describe the overall ctenophore propulsion system, we define nine morphometric and five kinematic parameters. These parameters are listed in Table 1, along with a brief description, while Fig 3 shows a graphical description of some parameters. Finally, Table 2 shows the average of the morphometric parameters as measured from eight studied individuals.

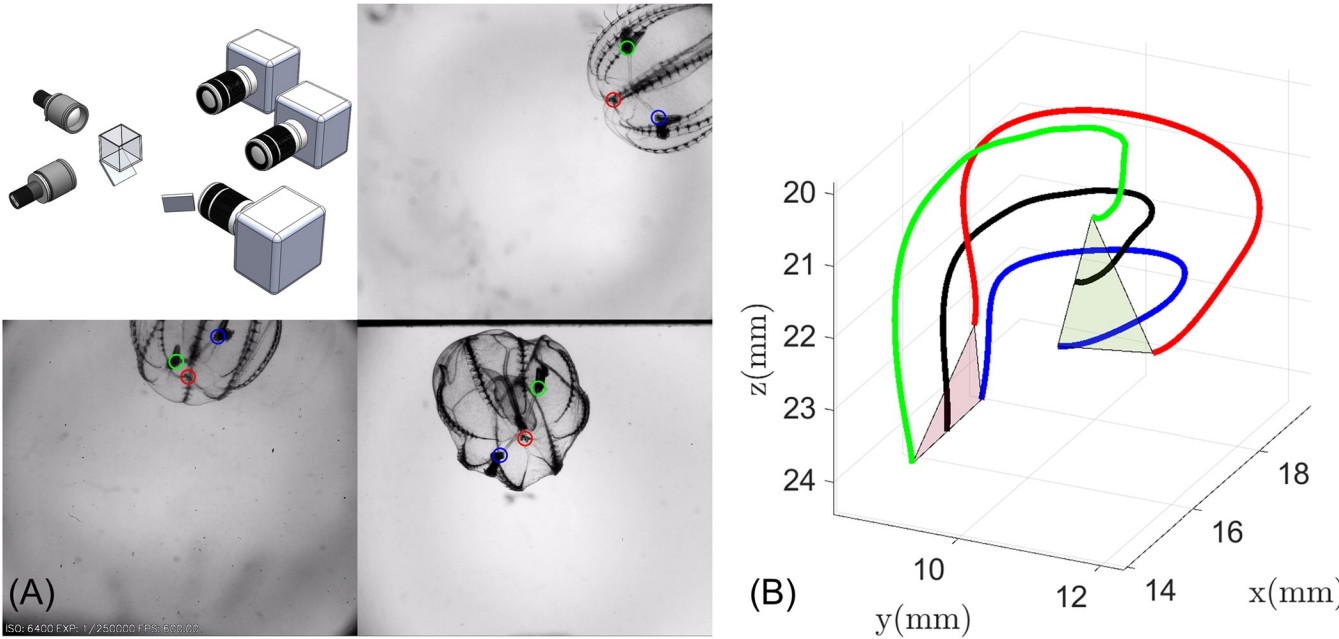

**Fig 2. Kinematic tracking experimental arena.** (A) Schematic of the 3D recording system showing the three orthogonal camera views. The three tracked points (apical organ (red) and tentacular bulbs (blue and green)), are shown in each camera view. (B) An example of a reconstructed trajectory; black line is the swimming trajectory, which we define to be the path of the midpoint of the line segment connecting the tentacular bulbs. Green and red triangles show the initial and final position of the animal.

## Ctenophore swimming model

Evaluating the full extent of the maneuverability and agility of an animal solely from behavioral observation is challenging; even with an extensive dataset, it is difficult to explore the parameter space in a systematic way (especially if there are parts of the parameter space the animal does not naturally occupy). Numerical methods (*e.g.*, computational fluid dynamics) are more

**Table 1. Ctenophore morphometric and kinematic parameters.**

| Variable | Description |
|---|---|
| $L_B$ | Body length |
| $d_B$ | Body diameter (measured in tentacular plane) |
| $l$ | Ctene length |
| $\delta$ | Average distance between ctenes |
| $n_S$ | Number of ctenes on each sagittal row (top and bottom rows in Fig 3A) |
| $n_T$ | Number of ctenes on each tentacular row (left and right rows in Fig 3A) |
| $\varepsilon_S$ | Sagittal ctene row position angles (measured from tentacular plane) |
| $\varepsilon_T$ | Tentacular ctene row position angles (measured from tentacular plane) |
| $\kappa$ | Position angle of the first ctene on the row (measured from centroid) |
| $f$ | Beat frequency |
| $\Phi$ | Stroke amplitude |
| $P_L$ | Phase lag between adjacent ctenes, expressed as a percentage of the cycle period |
| $Ta = \frac{t_r - t_p}{t_r + t_p}$ | Temporal asymmetry, quantifying the time difference between the power ($t_p$) and recovery strokes ($t_r$); also known as the "kinematic parameter" [31] |
| $Sa = \frac{A_e}{A_o}$ | Spatial asymmetry, quantifying the degree of difference in flow-normal area between the power and recovery stroke by comparing the area enclosed by the ctene tip trajectory $A_e$ to its practical maximum $A_o$ [27] |

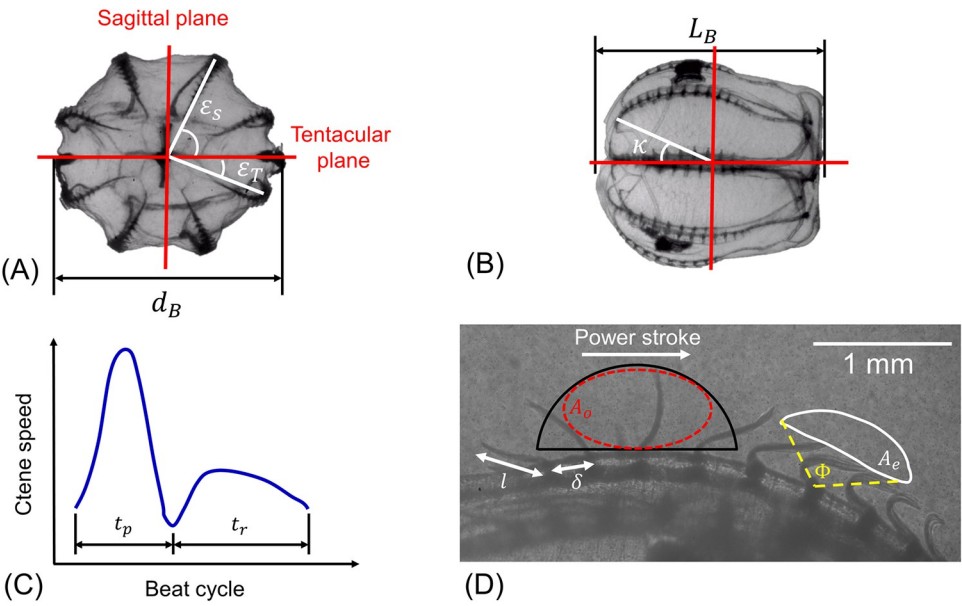

**Fig 3. Morphology and ctene row kinematics of a typical *Bolinopsis vitrea*.** (A) Top view showing the eight ctene rows, the ctene row position angle $\varepsilon$, and the sagittal and tentacular planes ($d_B = 7.6mm$); the "sagittal rows" are the rows adjacent to the sagittal plane, while the "tentacular rows" are adjacent to the tentacular plane. (B) Side view showing the ctene rows along the body ($L_B = 7.4mm$), and $\kappa$, the angle for the most aboral ctene. (C) Stylized example timeseries of ctene tip speed for one ctene over one beat cycle, where $t_p$ is the power stroke duration and $t_r$ the recovery stroke duration. (D) Ctene row close side view, showing a tracked ctene tip trajectory ($A_e$, solid white line), and the estimated ctene reachable space ($A_o$, red dashed ellipsoid inscribed in black half circle of radius $l$; shown elsewhere on the ctene row for clarity). Stroke amplitude ($\Phi$) and the direction of the power stroke are also marked.

controllable [32–34], but computational costs can be prohibitive for a parameter sweep of a highly multivariate problem. For this specific case, we also require a large domain to simulate an entire swimming maneuver (on the scale of centimeters) while resolving flow around the ctene rows (sub-millimeter scale). The ctenes are at least twenty times smaller than the body, so the computational resources needed for a fully-coupled model of even a few ctenes in a row are already a limiting factor [34–37]. However, a simplified modeling approach is still attractive due to the large and multivariate parameter space we seek to explore. We therefore develop a reduced-order analytical model based on known empirical expressions for fluid drag—an approach that has been previously used to study metachronal rowing in 1D for low and intermediate Reynolds numbers [1,27,38,39]. This class of analytical model is limited because it does not consider hydrodynamic interactions between the propulsors, and therefore cannot fully reproduce key features (such as enhanced swimming efficiency) of metachronal swimming. However, it can still reasonably predict swimming kinematics, and (most importantly) it provides a useful tool for comparing the relative effects of the many morphometric and kinematic parameters involved in metachronal swimming without prohibitive computational cost.

In this section, we expand the 1D formulation found in [27] to three dimensions, and use it to study ctenophore maneuverability. Unlike several similar models, here we fully incorporate the combination of viscous and inertial effects which arises at intermediate Reynolds numbers

**Table 2. Morphometric measurements of included *B. vitrea* (mean ± one standard deviation).**

| $L_B(mm)$ | $d_B(mm)$ | $l(mm)$ | $s$ | $n_S$ | $n_T$ | $\varepsilon_S(°)$ | $\varepsilon_T(°)$ | $\kappa(°)$ |
|---|---|---|---|---|---|---|---|---|
| 7.8±1.6 | 6.1±1.7 | 0.5±0.06 | 0.8±0.2 | 10±1.7 | 7.1±1.2 | 63.9±2.1 | 23±2.4 | 27±5.1 |

**Table 3. Reduced-order swimming model parameters.** Vector quantities are expressed in the global frame unless marked with a prime (as in $\vec{\omega}'$).

| Variable | Description |
|---|---|
| $\psi, \theta, \phi$ | Euler angles (yaw, pitch, and roll) |
| $\vec{F}_{net}$ | Net propulsive force |
| $\vec{F}_D$ | Body drag |
| $\vec{F}_{AR}$ | Acceleration reaction force |
| $\vec{X}_{B/O}$ | Body position vector |
| $m$ | Body mass |
| $\vec{T}'_{net}$ | Net propulsion torque |
| $\vec{T}'_{op}$ | Opposing torque |
| $\vec{\omega}'$ | Angular velocity vector |
| $I$ | Moment of inertia matrix |
| $R$ | Rotation matrix |
| $w$ | Plate width |
| $y_A$ | Instantaneous plate length |
| $x_A$ | Instantaneous plate oscillatory position |
| $C_A$ | Plate drag coefficient |
| $\vec{u}$ | Plate instantaneous velocity vector |
| $C_B$ | Body drag coefficient |
| $a$ | Body semi-minor axis |
| $b$ | Body semi-major axis |
| $\vec{r}'$ | Ctene position vector |
| $\tau$ | Phase lag time |
| $C_m$ | Added mass coefficient |
| $C_R$ | Torque coefficient |

by ensuring that relevant drag and torque coefficients are a function of the instantaneous speed and geometry of both the body and the ctenes. Based on the average body and appendage length (Table 2), the maximum swimming speed (2.7 $BL/s$) and maximum beat frequency (34 $Hz$), we calculate body and appendage-based Reynolds numbers of 157 and 57 ($Re_b = UL/v$, $and$ $Re_\omega = 2\pi f l^2/v$).

We model the ctenophore as a self-propelled spheroidal body suspended in a quasi-static flow, whose motion is governed by the balance between the propulsive and opposing forces and torques. Table 3 lists all the model parameters. To describe the motion of the spheroidal body, we require two coordinate systems: a global (fixed) coordinate system, in which a vector is expressed as $\vec{x} = x_1\hat{e}_1 + x_2\hat{e}_2 + x_3\hat{e}_3$, and a body-based coordinate system in which $\vec{x}' = x_1'\hat{e}_1' + x_2'\hat{e}_2' + x_3'\hat{e}_3'$ (see Fig 4).

As is typical in vehicle dynamics [40] we relate the orientation between both coordinate systems by successive rotations: yaw ($\psi$, rotation about $\hat{e}_3'$), pitch, ($\theta$, rotation about $\hat{e}_2'$) and roll ($\phi$, rotation about $\hat{e}_1'$). The transformation between the global and body frames is given by $\vec{x}' = R\vec{x}$, where the transformation (rotation) matrix is given by

$$R = \begin{bmatrix} c(\theta)c(\psi) & c(\theta)s(\psi) & -s(\theta) \\ s(\phi)s(\theta)c(\psi) - c(\phi)s(\psi) & s(\phi)s(\theta)s(\psi) + c(\phi)c(\psi) & s(\phi)c(\theta) \\ c(\phi)s(\theta)c(\psi) + s(\phi)s(\psi) & c(\phi)s(\theta)s(\psi) - s(\phi)c(\psi) & c(\phi)c(\theta) \end{bmatrix} \quad (1)$$

where ($\psi, \theta, \phi$) are the Euler angles, and c($\cdot$) and s($\cdot$) denote cosine and sine, respectively.

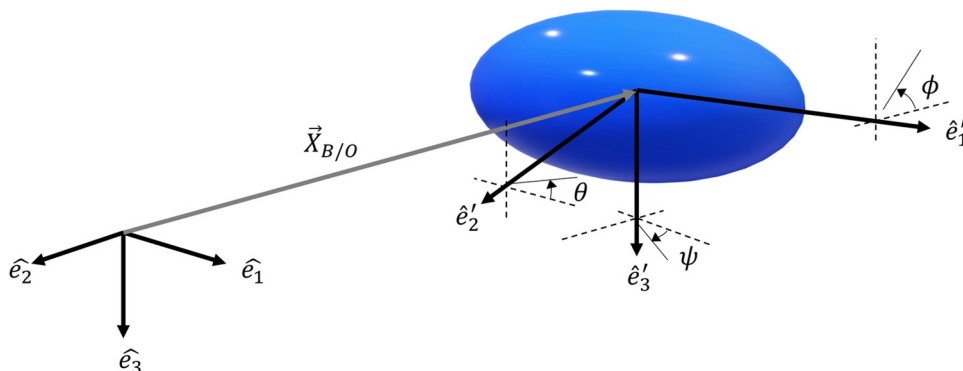

**Fig 4. Schematic of a ctenophore's simplified geometry moving in 3D space.** The unit vectors $\hat{e}_1$, $\hat{e}_2$, and $\hat{e}_3$ define the global (fixed) coordinate system while $\hat{e}'_1$, $\hat{e}'_2$, and $\hat{e}'_3$ correspond to the moving coordinate system attached to the spheroidal body.

The differential equations describing this balance are based on Euler's first and second laws (Eqs 2 and 3). Eq 2 balances the propulsive force ($\overrightarrow{F}_{net}$), the drag force ($\overrightarrow{F}_D$), the acceleration reaction force ($\overrightarrow{F}_{AR}$), the body mass ($m$), and the body acceleration with respect to the origin ($\ddot{\overrightarrow{X}}_{B/O}$). Eq 3 balances the propulsive torque ($\overrightarrow{T}'_{net}$) and the opposing torque ($\overrightarrow{T}'_{op}$) with the moment of inertia matrix $[I]$ and the body's angular velocity ($\overrightarrow{\omega}'$) and acceleration ($\dot{\overrightarrow{\omega}}'$). We will define each one of these terms in the following subsections. However, we direct the reader to the supplementary material S1 Text for details of the solution procedure, the numerical implementation, the formulations for various coefficients, and the validation of the model against experimental data.

$$\overrightarrow{F}_{net} + \overrightarrow{F}_D + \overrightarrow{F}_{AR} = m\ddot{\overrightarrow{X}}_{B/O} \tag{2}$$

$$\overrightarrow{T}'_{net} + \overrightarrow{T}'_{op} = [I]\dot{\overrightarrow{\omega}}' + \overrightarrow{\omega}' \times ([I]\overrightarrow{\omega}') \tag{3}$$

## Expressions for propulsive forces and torques

As seen in Fig 5B, the ctene tip follows a roughly elliptical trajectory during the power-recovery cycle. During the power stroke, the paddle is extended and moving quickly; during the recovery stroke, the paddle is bent and moving slowly. Such a cycle is both spatially asymmetric (higher flow-normal area on the power stroke *vs.* the recovery stroke) and temporally asymmetric (power stroke duration shorter than recovery stroke duration). To model this, we consider each ctene as an oscillating flat plate with a time-varying length, whose proximal end oscillates along a plane tangent to the body surface and whose distal end traces an ellipse (Fig 5D). The coordinates of the distal end ($x_A(t)$, $y_A(t)$) are prescribed parametrically, and are constrained by five parameters: the maximum length of the ctene ($l$), the stroke amplitude ($\Phi$), the beat frequency ($f$), the temporal asymmetry ($Ta$), and the spatial asymmetry ($Sa$). Further details are found in the supplementary material (S1 Text). We chose to model the ctenes as oscillating flat plates in part due to the availability of drag coefficient empirical expressions for intermediate Reynolds numbers ($1<Re_\omega<1000$) [41], as the non-independence of drag from $Re_\omega$ is a key feature of intermediate-$Re$ swimming. The oscillating plate model also represents

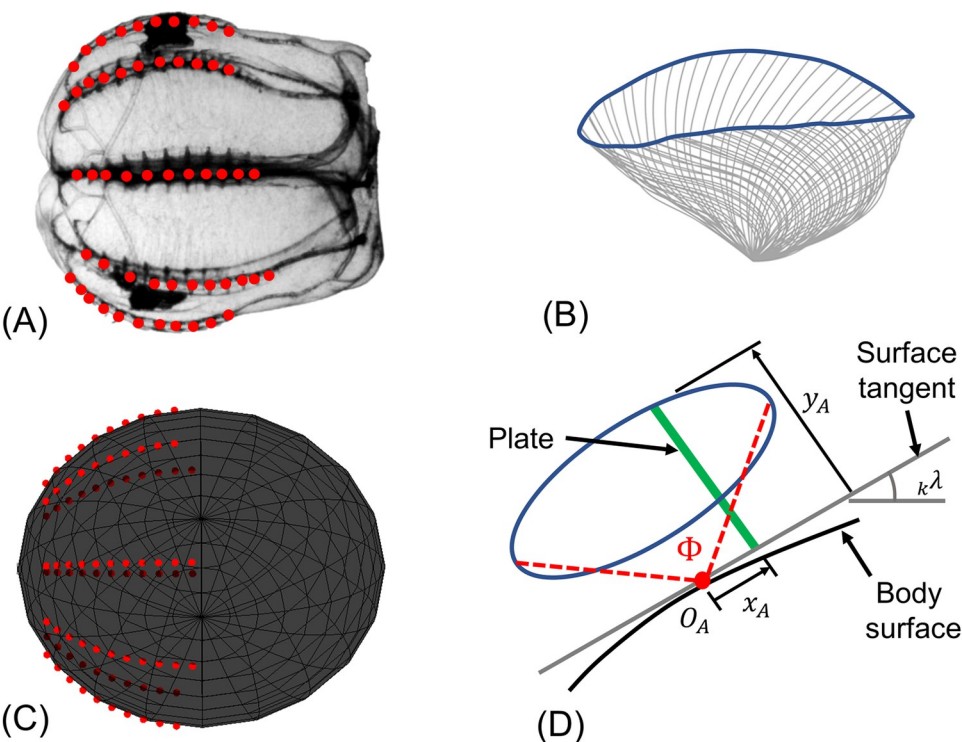

**Fig 5. Ctenophore reduced-order modeling.** (A) Lateral view of a ctenophore; red dots mark the position of the ctenes that circumscribe its body in eight rows. (B) Real ctene tip trajectory from a tracked time series of ctene kinematics (gray lines, spaced equally in time). (C) Ctenophore modeled as a spheroidal body; red dots indicate the application point for each modeled (time-varying) ctene propulsion force. (D) Simplified elliptical trajectory for a modeled ctene, which is a flat plate with time-varying length. The plate oscillates parallel to a plane tangent to the curved surface of the modeled body ($_k\lambda$, tangential angle to the body surface). The time-varying tip position ($x_A$, $y_A$) is prescribed as a function of the five ctene beating control parameters: $f$, $\Phi$, $l$, $Sa$, and $Ta$ (see S1 Video).

arguably the simplest and most generalizable case of spatially asymmetric rowing, changing only the flow-normal area (and not the paddle angle or other variables) between the power and the recovery stroke.

We "place" a modeled ctene in each of the ctene positions (determined by $\varepsilon$, $\delta$, and $\kappa$, coupled with the body geometry) around the spheroidal body (Fig 5A and 5C). Each ctene oscillates around its initial position (Fig 5D), creating a force tangential to the body surface (S1 Video). The total propulsive force of the $i^{th}$ ctene row is modeled as the negative of the drag force summed over each of $n$ oscillating plates:

$$_i\overrightarrow{F} = -\frac{\rho w}{2}\sum_{k=1}^{n} {}_{ik}y_A \; {}_{ik}C_A\left[|\dot{\overrightarrow{X}}_{B/O} + {}_{ik}\overrightarrow{u}|^2 \frac{{}_{ik}\overrightarrow{u}}{|{}_{ik}\overrightarrow{u}|}\right] \qquad (4)$$

where $\rho$ is the fluid density and $n$ is the number of ctenes in a given row ($k$ is the index of the ctene). The flow-normal area is given by the plate width $w$ (assumed to be 0.5·$l$ [10]) times the instantaneous plate length $_{ik}y_A(t+(k-1)\tau)$. The drag coefficient $_{ik}C_A$ is that of an oscillating flat plate at intermediate Reynolds number, and is a function of the instantaneous plate speed $\dot{x}_A$ [41]. The force is proportional to the square of the magnitude of the global ctene velocity vector $\dot{\overrightarrow{X}}_{B/O} + {}_{ik}\overrightarrow{u}$, where $\dot{\overrightarrow{X}}_{B/O}$ is the body velocity with respect to the origin and $_{ik}\overrightarrow{u}$ is the velocity of the $k^{th}$ plate in the $i^{th}$ row in the global frame, which is itself a function of the

instantaneous plate oscillatory speed $_{ik}\dot{x}_A(t + (k-1)\tau)$:

$$_{ik}\overrightarrow{u} = \boldsymbol{R}^T {}_{ik}\dot{x}_A[c(_k\lambda)\hat{e}'_1 + s(_k\lambda)c(_i\varepsilon)\hat{e}'_2 + s(_k\lambda)s(_i\varepsilon)\hat{e}'_2] \tag{5}$$

where $_k\lambda$ is the angle defining the tangent to the body surface at the $k^{\text{th}}$ plate (see Fig 5D). Metachronal coordination is incorporated by dephasing the plate kinematic variables $_{ik}\dot{x}_A$ and $_{ik}y_A$ by an amount $(k-1)\tau$, where $\tau = P_L \cdot T$. Considering all $N$ ctene rows, the net propulsive force is

$$\overrightarrow{F}_{net} = \sum_{i=1}^{N} {}_i\overrightarrow{F} \tag{6}$$

We note that a one-dimensional model of a single ctene row is described in [27] and the sensitivity of the net force to various model inputs is explored therein.

Propulsive torque is calculated as the cross product of the ctene's position relative to the centroid of the body and the force generated by the ctene:

$$\overrightarrow{T}'_{net} = \sum_{i=1}^{N} \sum_{k=1}^{n} {}_{ik}\overrightarrow{r}' \times -\frac{\rho w}{2}\mathbf{R}\left[{}_{ik}y_{Aik}C_A\left(|\dot{\overrightarrow{X}}_{B/O} + {}_{ik}\overrightarrow{u}|^2 \frac{_{ik}\overrightarrow{u}}{|_{ik}\overrightarrow{u}|}\right)\right] \tag{7}$$

where $_{ik}\overrightarrow{r}'$ is the position vector of the $k^{th}$ ctene in the $i^{th}$ row (relative to the body centroid), and the bracketed term is the ctene propulsion force in the global frame of reference. To calculate the propulsive torque, the propulsive force must be expressed in the body frame of reference; hence, we multiply it by the transformation matrix $\boldsymbol{R}$.

## Expressions for resistive forces and torques

The drag force on the 3D spheroidal body is:

$$\overrightarrow{F}_D = -\mathbf{R}^{\mathbf{T}}\frac{\rho}{2}\begin{bmatrix} (\pi a^2)C_B^{\parallel}((\mathbf{R}\dot{\overrightarrow{X}}_{B/O}|\dot{\overrightarrow{X}}_{B/O}|) \cdot \hat{e}'_1) \\ (\pi ab)C_B^{\perp}((\mathbf{R}\dot{\overrightarrow{X}}_{B/O}|\dot{\overrightarrow{X}}_{B/O}|) \cdot \hat{e}'_2) \\ (\pi ab)C_B^{\perp}((\mathbf{R}\dot{\overrightarrow{X}}_{B/O}|\dot{\overrightarrow{X}}_{B/O}|) \cdot \hat{e}'_3) \end{bmatrix} \tag{8}$$

Because the body is spheroidal, we must consider two drag coefficients: $C_B^{\parallel}$ is the drag coefficient for the longitudinal movements (roll axis, $\hat{e}'_1$), and $C_B^{\perp}$ is the drag coefficient for the lateral movements (pitch and yaw axes, $\hat{e}'_2 \& \hat{e}'_3$). Because we are in the viscous-inertial (intermediate Reynolds number) regime, $C_B^{\parallel}$ and $C_B^{\perp}$ are each a function of both speed and geometry [42]. These coefficients are multiplied by the respective velocity squared components (transformed to the body frame of reference by the transformation matrix $\boldsymbol{R}$), the corresponding flow normal area ($\pi a^2$, for $C_B^{\parallel}$, and $\pi ab$, for $C_B^{\perp}$), the fluid density, and a factor of 1/2. Finally, to transform the components of the drag force back to the global frame of reference, we multiply by the transpose of the transformation matrix $\mathbf{R}^{\mathbf{T}}$. The drag force on the ctenes has already been incorporated as part of $_i\overrightarrow{F}$, which opposes the direction of motion during the ctene's recovery stroke.

The acceleration reaction (added mass) force is calculated as $C_m\rho V$, where $\rho$ is the fluid density, $V$ is the body volume, and $C_m$ is the added mass coefficient, which depends on the body shape and the direction of motion [43]. We need two added mass coefficients for our spheroidal body: $C_m^{\parallel}$, for motion along to the roll axis, and $C_m^{\perp}$, for motion along to the pitch/yaw axes

[44]. Similar to the derivation of the drag force (Eq 8), we have:

$$\overrightarrow{F}_{AR} = -\boldsymbol{R}^{\mathrm{T}} \rho V \begin{bmatrix} C_m^{\parallel} (\mathbf{R}\overset{..}{\overrightarrow{X}}_{B/O} \cdot \hat{e}_1') \\ C_m^{\perp} (\mathbf{R}\overset{..}{\overrightarrow{X}}_{B/O} \cdot \hat{e}_2') \\ C_m^{\perp} (\mathbf{R}\overset{..}{\overrightarrow{X}}_{B/O} \cdot \hat{e}_3') \end{bmatrix} \tag{9}$$

Finally, we model the overall resistance to body rotation, notated as the opposing torque ($\overrightarrow{T}'_{op}$). The opposing torque comes from both viscous drag and acceleration reaction forces; however, an analytical formulation of this torque is outside the scope of this model. Here we use an expression based on torque coefficients for rotating prolate spheroids at intermediate Reynolds numbers, which are taken from numerical simulations [45]:

$$\overrightarrow{T}'_{op} = -\frac{\rho}{2}\left(\frac{d_e}{2}\right)^5 \begin{bmatrix} sgn(\omega_x')C_R^{\parallel}\omega_{x'}^2 \\ sgn(\omega_y')C_R^{\perp}\omega_{y'}^2 \\ sgn(\omega_z')C_R^{\perp}\omega_{z'}^2 \end{bmatrix} \tag{10}$$

where $d_e$ is the equivalent sphere diameter (i.e., the diameter of a sphere with the same volume as the spheroid), $C_R^{\parallel}$ is the torque coefficient for rolling, and $C_R^{\perp}$ for pitch and yaw. Both coefficients are a function of angular speed and geometry (see S1 Text). The sign function is introduced so that the resistive torque always opposes the body motion.

## Model performance and validation

We use our reduced-order model to create simulated trajectories of freely swimming ctenophores to explore the omnidirectional capability and general turning performance of all the possible turning strategies of the ctenophore locomotor system across the available parameter space. The morphometric parameters for the model are based on the mean values of our experimental observations (Table 2). The kinematic parameters (stroke amplitude $\Phi$, phase-lag $P_L$, temporal asymmetry $Ta$, and spatial asymmetry $Sa$) cannot be directly measured from the video recordings of the swimming trajectories. However, kinematic parameters for the same set of animals, filmed at a higher spatial resolution, are reported in [27]. Based on these measurements, we use the following representative values: $\Phi = 112°$, $P_L = 13.2\%$, and (based on the model validation, see S1 Text) $Ta = 0.3$, and $Sa = 0.3$.

We validate the model against experimentally measured trajectories to ensure that it is able to capture key features (see S1 Text). Despite its limitations (the model does not capture hydrodynamic interactions, nor does it account for the flexibility and deformability present in the biological ctenes), we reproduce large-scale features of the trajectories. When the model is prescribed to replicate the measured time-varying frequencies observed in freely swimming animals, given the same starting point, the mean radius of curvature and mean swimming speed (our primary variables of interest) vary by no more than 17% compared to the measured trajectory. However, small-scale features of the trajectory may not be reproduced. This is not our goal; instead, we aim to reproduce these large-scale features to quickly sweep the parameter space in a way that would be computationally unfeasible with a more faithful model which captured hydrodynamic interactions and other variables which certainly affect the trajectory.

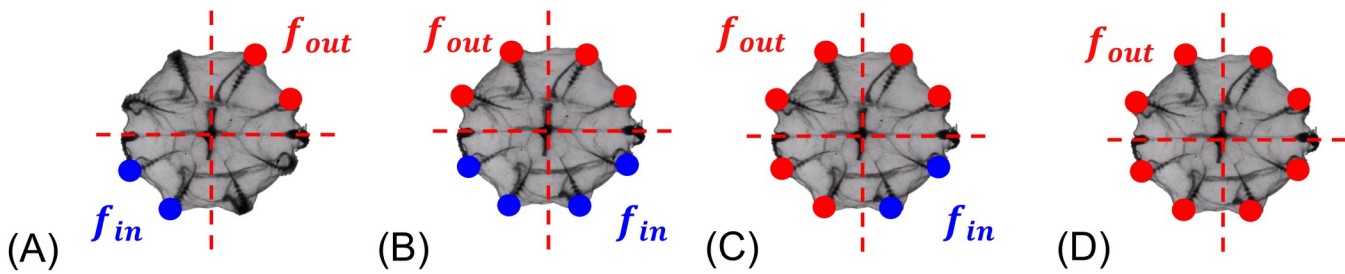

**Fig 6. Ctenophores are capable of several different swimming control strategies based on the activation of their ctene rows, which are controlled in pairs (such that each body quadrant receives one frequency input).** While each quadrant can operate at an independent frequency, we typically observe only two frequencies during turns, such that there is a single frequency differential ($f_{out}$>$f_{in}$). The above images schematically show (A) turning mode 1, (B) turning mode 2, (C) turning mode 3, and (D) straight swimming (mode 4).

## Results

### Turning performance

From 27 recorded sequences, we observed four different appendage control strategies. These strategies differ categorically in the total number and the geometrical arrangement of the rows actively beating. The first three strategies are used to turn, with rows on the outside of the turn beating at a higher frequency than the rows on the inside of the turn ($f_{out}$>$f_{in}$). In the first strategy (mode 1), two adjacent rows beat at some frequency $f_{out}$ and the two opposite rows beat at a lower frequency $f_{in}$ while the remaining four rows are inactive. In the second strategy (mode 2), the four outer rows beat at approximately the same frequency, which exceeds the frequency used by the four rows on the opposite side. For the third strategy (mode 3), six rows beat at a constant frequency $f_{out}$ while only two rows beat at a lower frequency $f_{in}$. Lastly, in mode 4 all rows are beating at approximately the same frequency; thus, the animal swims roughly in a straight line. Fig 6 shows a schematic depiction of the activated ctene rows for each control strategy. The observed control strategies agree with morphological studies of lobate ctenophores [18]: the apical organ has four compound balancer cilia, and each balancer controls the activation of one sagittal and one tentacular row. In other words, *B. vitrea* can independently control the ctenes in each of the body quadrants formed by the sagittal and tentacular planes (see Fig 3A), but the two rows in each quadrant beat at approximately the same frequency. Table 4 shows a summary of the control strategies and the number of times each was observed. The recorded beat frequencies range from 0 to 34.5 *Hz*. Examples of the four strategies can be seen in S2–S5 Videos.

To explore the turning performance of *B. vitrea*, we use the observed 3D swimming trajectories and the mathematical model to build a maneuverability-agility plot (MAP). In Fig 7, the x-axis shows the average animal speed during the recorded sequence ($\overline{V}$), measured in body lengths per second, which we treat as a measure of agility. The y-axis shows the average normalized radius of curvature ($\overline{R/L}$), which we treat as a measure of maneuverability.

**Table 4. Appendage control strategies observed in freely swimming *B. vitrea* (27 total trajectories).**

| Control strategy | No. of rows beating at $f_{out}$ | No. of rows beating at $f_{in}$ | No. of observations |
|---|---|---|---|
| Mode 1 | 2 | 2 | 2 |
| Mode 2 | 4 | 4 | 8 |
| Mode 3 | 6 | 2 | 9 |
| Mode 4 | 8 | 0 | 8 |

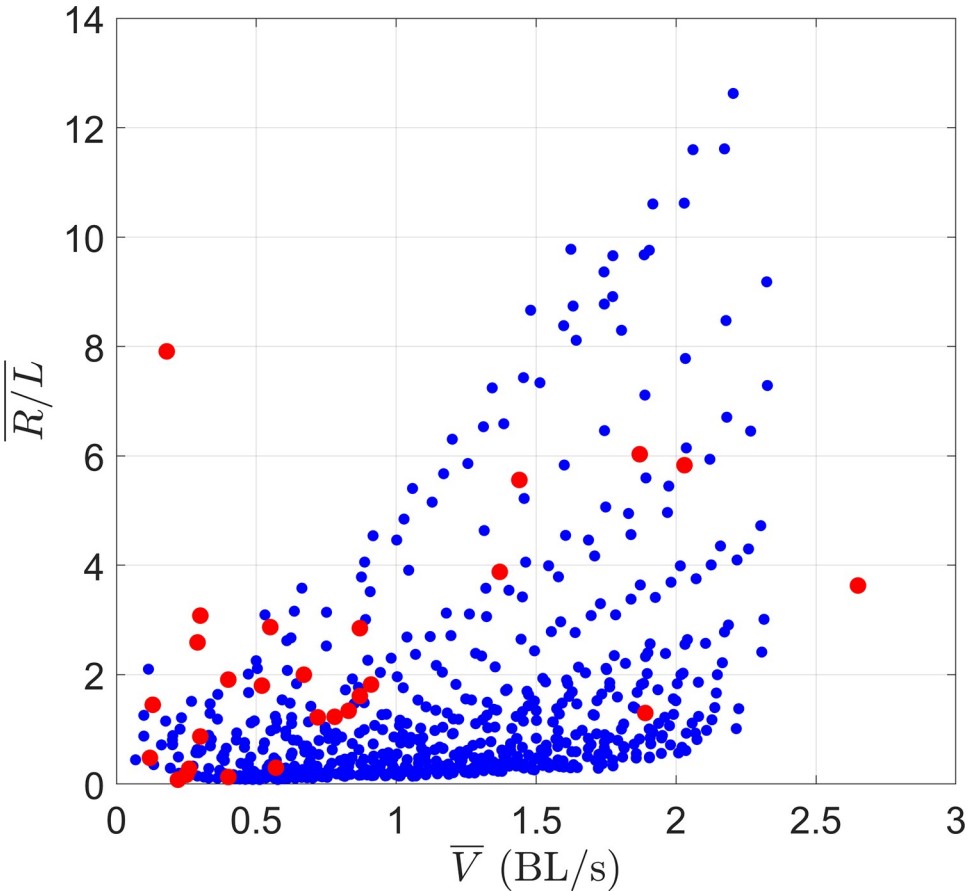

**Fig 7. Maneuverability-Agility Plot (MAP).** Experimental measurements of freely swimming *B. vitrea* (red dots) and for all simulated cases of modes 1, 2, and 3 (blue dots). Lower values of $\overline{R/L}$ indicate sharp turns (more maneuverable); higher values of $\overline{V}$ indicate faster swimming (more agile). Values in the upper left (low $\overline{V}$, high $\overline{R/L}$) are straightforwardly achievable with straight swimming (mode 4) or with $\Delta f < 2Hz$; these points were not simulated. Simulating mode 4 mathematically would result in $\overline{R/L} \sim \infty$, since the eight rows beat at the same frequency. However, mode 3 will approach the behavior of mode 4 as $\Delta f = f_{out} - f_{in}$ approaches zero. Here, the minimum value is $\Delta f = 2Hz$, so the upper-left corner of the MAP is not occupied. Simulations were halted after the timestep in which $\overline{R/L}$ exceeded 10, resulting in some trials with $\overline{R/L}$ slightly greater than 10.

Movements that are both highly maneuverable and highly agile are found in the lower-right corner of the MAP, while highly maneuverable but less agile (slow) movements are found in the lower-left corner. From the experimental observations (red dots), we observe an increase in $\overline{R/L}$ as $\overline{V}$ increases, an expected tradeoff between maneuverability and agility [24]. The most maneuverable observed turn has $\overline{R/L} = 0.08$ at a speed of $\overline{V} = 0.22\ BL/s$ (lower-left corner). In the lower-right corner, we have a turn with a measured speed of $\overline{V} = 1.89\ BL/s$ for $\overline{R/L} = 1.3$—still a comparatively sharp turn, carried out at 71% of the maximum recorded straight-line swimming speed of $\overline{V} = 2.65\ BL/s$ (rightmost point, Fig 7). For context, a much faster centimeter-scale swimmer, the whirligig beetle ($L_B = 12.38\ mm$, $V_{max} = 44.5\ BL/s$), can turn at $\overline{V} = 22.42\ BL/s$ for $\overline{R/L} = 0.86$; this is 50.4% of their maximum observed speed [24]. While ctenophores typically swim at a much lower (normalized) speed, they are capable of sharp turning while maintaining nontrivial speeds—that is, they have both high maneuverability and high agility.

**Table 5. Range and resolution of the frequencies used in the analytical simulations.**

|  | $f_{out}$(Hz) | $f_{in}$(Hz) |
|---|---|---|
| Range | 2–34 | $0 — (f_{out} - 2\ Hz)$ |
| Resolution | 2 | 2 |

We use the mathematical model to expand our analysis of *B. vitrea's* turning performance by simulating all possible configurations of modes 1, 2, and 3. We ran a total of 612 simulations covering the range and resolution of the beat frequencies reported in Table 5. Each simulation continued until the average of the normalized radius of curvature $(\overline{R/L})$ over two seconds (simulation time) reached steady-state, or if $\overline{R/L}$ exceeded 10, which we considered straight swimming. Fig 7 shows the simulated range (blue dots) with the experimental results (red dots). Our model predicts that *B. vitrea's* locomotor system can reach $\overline{R/L} = 0.08$ at a speed of $\overline{V} = 0.58\ BL/s$ (lower-left corner of the MAP, maximizing maneuverability). However, the system is also capable of significant maneuverability at high speeds: in the lower-right corner of the MAP (highly maneuverable and agile), the system can reach a speed of $\overline{V} = 2.33\ BL/s$ for $\overline{R/L} = 0.98$. These two data points range from 24% to 93% of the simulated top speed ($V_{max} = 2.49\ BL/s$, with eight rows beating at 34 Hz), while still maintaining a turning radius of less than one body length. The model results confirm that ctenophores' metachronal rowing platform is highly maneuverable and agile, with performance limits that may extend beyond our experimental observations.

We can use the overall MAP (Fig 7) to further clarify the differences between turning modes. Fig 8 shows the MAP for each individual turning mode along with the beat frequency differential between the rows on the outside and inside of the turn ($\Delta f = f_{out} - f_{in}$). As expected, increasing $\Delta f$ results in high maneuverability (smaller $\overline{R/L}$); the model results are incremented from $f_{out} = 2\ Hz$ and $f_{in} = 0\ Hz$ to $f_{out} = 34\ Hz$ and $f_{in} = 32\ Hz$.

## Omnidirectionality

Using the observed 3D swimming trajectories, we estimate *B. vitrea's* motor volume (MV) [26], which illustrates the maneuvering capabilities of the ctenophore locomotor system (Fig 9). Conceptually, the MV represents the reachable space of a swimming ctenophore over a given time horizon. To build the MV, we translated and rotated the observed swimming trajectories so that (at the start of the trajectory) the tentacular plane is aligned with the x-y plane, the midpoint between the tentacular bulbs is at the origin, and the aboral-oral axis of symmetry is aligned with the x-axis with the oral end facing the positive x-direction (see yellow sketches in Fig 9). From this starting position, the positive x-direction is forward swimming (lobes in front) and the negative x-direction is backward swimming (apical organ in front). Fig 9 shows the rearranged swimming trajectories (black lines) and the volume swept by the animals' bodies (gray cloud). Each animal body was estimated as a prolate spheroid based on its unique body length and diameter ($L_B$, $d_B$). In our observations, animals swam freely (without external stimuli), and the trajectories were recorded through the time period that the animal was in the field of view. Therefore, each observation has a different initial speed and total swimming time (see Table 6). This is therefore not a direct comparison of different appendage control strategies, since observed maneuvers have different initial speeds and durations. We also note that because we only observed animals who freely swam through the field of view, the dataset is biased towards animals who had a nontrivial initial swimming speed, leading to a stretching of the MV along the x-axis. Nonetheless, the observed MV shown in Fig 9 provides some visualization of the 3D maneuvering capabilities of *B. vitrea's* locomotor system.

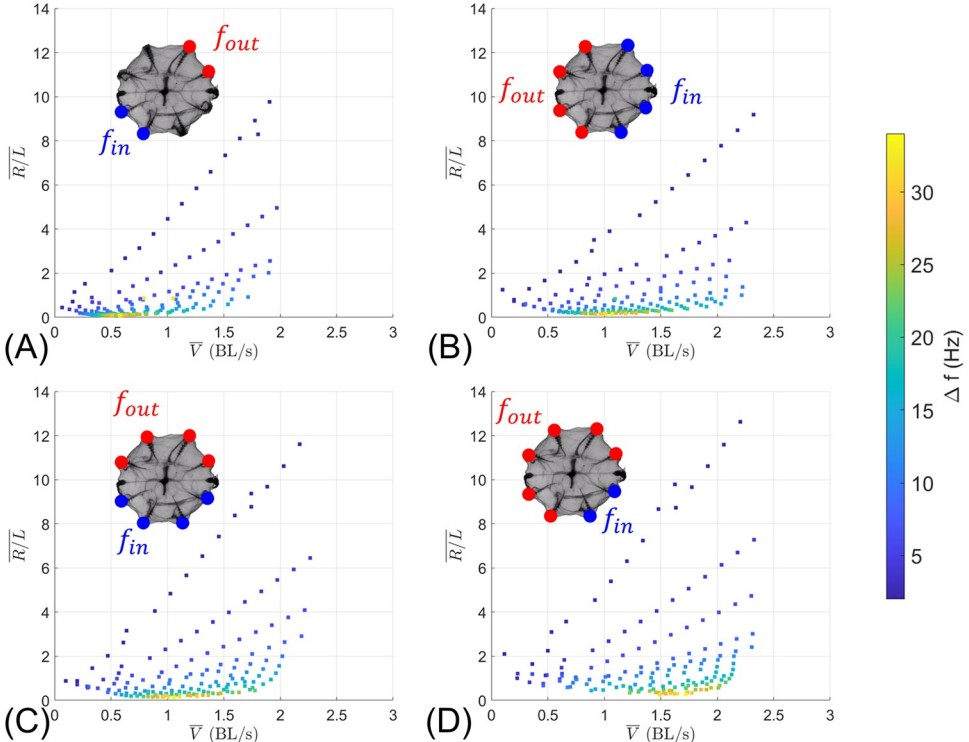

**Fig 8. MAPs for the different turning modes, along with beat frequency differential ($\Delta f = f_{out} - f_{in}$).** (A) Turning mode 1 (2 vs. 2 ctene rows). (B) Turning mode 2 (4 vs. 4 ctene rows) with consecutive sagittal rows at different frequencies. (C) Turning mode 2 (4 vs. 4 ctene rows) with consecutive tentacular rows at different frequencies. (D) Turning mode 3 (6 vs. 2 ctene rows).

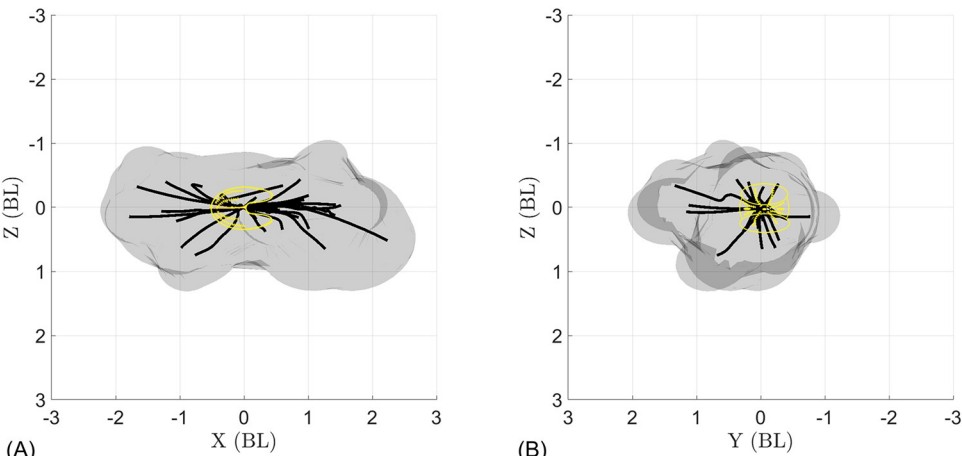

**Fig 9. Motor volume (MV) constructed from the 27 tracked swimming trajectories of *B. vitrea*.** Black lines show swimming trajectories (midpoint between tentacular bulbs) and volume swept by animals' bodies (gray cloud) during each maneuver. Animal volume is estimated as a prolate spheroid based on morphological measurements (Table 2). The yellow sketches indicate the initial position of the ctenophore; the motor volume is elongated in X because all trajectories were considered from the same starting orientation, and because our dataset contains only animals with a nonzero initial velocity (since animals freely swam through the field of view). (A) Side view and (B) front view of the tracked swimming trajectories and motor volume show that *B. vitrea* can turn over a large range of angles.

**Table 6. Experimental recordings (mean ± one standard deviation).**

| Swimming direction | No. Recordings | Initial speed ($BL/s$) | Recording duration ($s$) |
|---|---|---|---|
| Forward (+x) | 19 | 0.61±0.75 | 2.25±1.34 |
| Backward (-x) | 8 | 0.85±0.55 | 1.82±1.11 |

Fig 9 shows the potential of the ctenophore locomotor system for omnidirectional swimming, which we define as the ability to move in any direction from a given initial position within a relatively small space and short time. Fig 9A shows nearly equal capacity between backward and forward swimming—an ability few swimmers share, and which typically requires major adjustments to control strategy [46]. Ctenophores, by contrast, achieve agile backward swimming simply by reversing the direction of the ctene power stroke (see S6 Video). The trajectories in Fig 9 are achieved via the activation of different ctene rows, which (when coupled with the ability to swim both forward and backward) allow ctenophores to quickly access many different swimming directions from the same initial position.

To explore the omnidirectional capabilities of *B. vitrea* in a more systematic fashion, we use the mathematical model to explore all possible permutations of modes 1, 2, and 3. For simplicity and clarity, Fig 10 displays only trajectories produced by active rows beating at a frequency of $f_{out}$ = 30 *Hz* and a $\Delta f$ = 30 *Hz* (so that all other rows are not active), for a simulation time of one second. As expected by the number of active rows, mode 1 is the most maneuverable of the three (shortest trajectories, Fig 10A); while mode 2 and mode 3 reach higher speeds while turning (longer trajectories, Fig 10A). This suggests that activating only two ctene rows (mode 1) could be best suited for fine orientation control (for example, when maintaining a vertical orientation when resting/feeding) [18]. The higher number of active appendages used in modes 2 or 3 could be used for escaping, where both high speed and rapid reorientation are needed [20]. Table 7 shows the maneuverability and agility levels of each mode simulated in

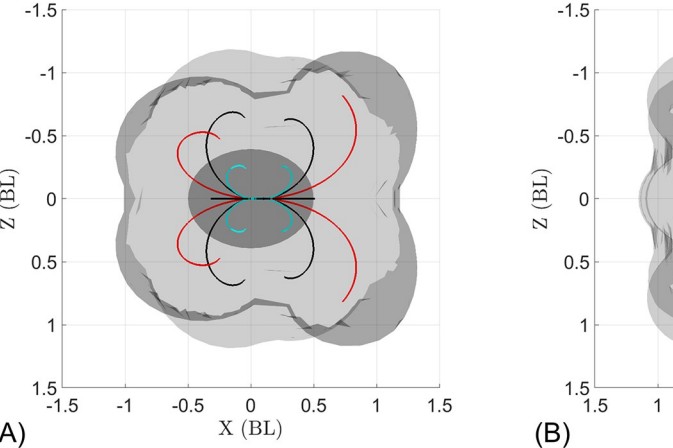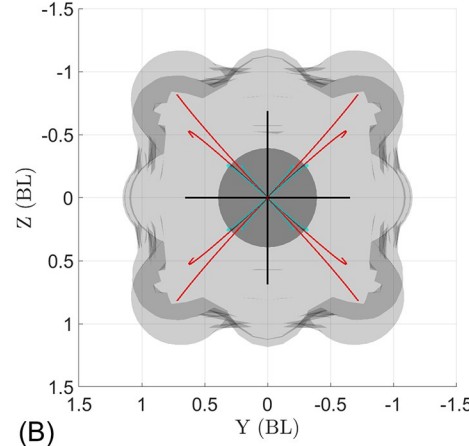

**Fig 10. Computationally simulated MV for the 3 ctenophore row control strategies, with a variable number of rows beating at 30 *Hz*, swimming either forward or backward, for a simulated time of one second.** The darker gray ellipsoid placed on the origin illustrates the animals' initial position. Turning mode 1 is shown in blue, mode 2 in black and mode 3 in red. (A) Side view displaying the backward (-x) and forward (+x) swimming trajectories. Asymmetry arises from the distribution of ctenes along the body. (B) Front view of the swimming trajectories, showing the wide range of turning directions.

**Table 7. Maneuverability and agility measurements for the simulated motor volume of a lobate ctenophore.**

| | Mode 1 | | Mode 2 | | Mode 3 | |
|---|---|---|---|---|---|---|
| | $\overline{R/L}$ | $\overline{V}(BL/s)$ | $\overline{R/L}$ | $\overline{V}(BL/s)$ | $\overline{R/L}$ | $\overline{V}(BL/s)$ |
| Forward (+x) | 0.20 | 0.60 | 0.34 | 1.09 | 0.72 | 1.62 |
| Backward (-x) | 0.19 | 0.60 | 0.31 | 1.05 | 0.47 | 1.39 |

Fig 10; we observe increasing values of $\overline{R/L}$ and $\overline{V}$ as we increase the number of active rows (modes 1 to 3). From Fig 10A it is also noticeable that backward swimming produces sharper turns than forward swimming. The value of $\overline{R/L}$ decreases (sharper turns) for backward swimming (see Table 7), and the discrepancy is more noticeable as we increase the number of active ctene rows (mode 1 to 3).

A front view of all modes (that is, the y-z plane) displays the range of swimming directions which are accessible from a given initial position (Fig 10B). This MV—which captures only a fraction the full capability of the swimming platform—shows the omnidirectionality of the ctenophore metachronal locomotor system, achieved only by constant pitching and yawing. In an actual swimming trajectory, a ctenophore can change the active rows, the frequency, or the turning mode over time, resulting in much more complex maneuvers (as in Fig 2B).

To fully explore the maneuvering capabilities of the ctenophore body plan, we examine the hypothetical case in which there is independent control of each ctene row. Fig 11 shows the MV for all the 255 non-repeatable permutations of activating $n_{cr}$ ctene rows at a time ($n_{cr}$ = 1,2,...,8) at 30 $Hz$ for a simulation time of one second. This MV shows that nearly any swimming direction can be accessed from the same initial position.

## Discussion and conclusions

Through a combination of freely swimming animal observations and a reduced-order analytical model, we have shown that metachronal swimming, particularly as used in the ctenophore body plan, represents significant untapped potential for bioinspired swimming robots. Ctenophores' higher number of propulsive rows differentiates them from other metachronal

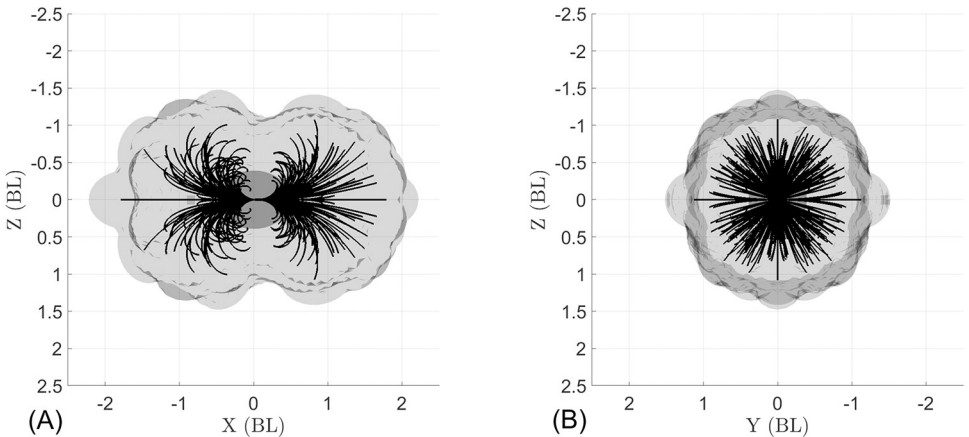

**Fig 11. Computationally simulated MV for 255 ctene row control strategies, with $1 \leq n_{cr} \leq 8$ rows beating at 30 $Hz$, swimming either forward or backward for a simulated time of one second.** (A) Side view displaying the backward (-x) and forward (+x) swimming trajectories. (B) Front view of the swimming trajectories, showing the wide range of turning directions. Across all simulated trajectories, the minimum achieved $\overline{R/L}$ is 0.22 (backward swimming) and 0.20 (forward swimming); the maximum achieved $\overline{V}$ is 1.79$BL/s$ (equal for forward and backward swimming).

swimmers, which typically have only one or two rows of propulsors [7,46–49]. Flexibility in controlling a higher number of appendages, combined with the ability to swim both backward and forward, allows for nearly omnidirectional swimming.

For the body plan studied here, which is typical of lobate ctenophores, we found that the asymmetric placement of ctenes within each row (i.e., ctenes distributed closer to the aboral than the oral end) enabled sharper turns during backward swimming when compared to forward swimming (Fig 10A). Ctene row asymmetries between the sagittal and tentacular rows of *B. vitrea* are due to the presence of the lobes (see Fig 1), which are used to create highly efficient feeding currents [50]. However, cydippid ctenophores such as *Pleurobrachia* sp. feed by capturing prey with their tentacles, then bringing the prey to their mouth by rotating their bodies [18]. In *Pleurobrachia* and other cydippids, ctenes are approximately symmetrically arranged from the oral to aboral end, which may eliminate the trajectory asymmetries observed in lobate ctenophores. It is likely that cydippid ctenophore swimming may be even more omnidirectional. To accomplish their stereotypical rotating behavior, cydippid ctenophores also reverse the direction of the power stroke on the inner ctene rows, potentially leading to even tighter turns that are not captured in our model. Another lobate ctenophore genus, *Ocyropsis*, contracts its lobes (like the bell of a jellyfish medusa) to increase its escape velocity, while still using ctene rows for orientation [21]; this indicates that ctene rows can be coupled with other propulsive strategies to achieve goals beyond that of maximizing maneuverability (*e.g.* to increase overall swimming speed). Extinct ctenophores had as many as 80 ctene rows, increasing the number of reachable turning planes. Some even had ctene rows placed diagonally on the body, potentially giving them the ability to roll [51]. Real ctenophores also use sporadic, irregular beating for fine-scale positional control, which is not captured in our model; this likely increases maneuverability even beyond what we have predicted.

Our results illustrate that the ctenophore body plan is highly agile and maneuverable, with the ability to turn sharply without slowing down, reverse directions easily, and turn about many planes, which enables them to access a nearly-unconstrained region of space from a given initial position over relatively short time horizons. We also find that different turning strategies occupy different regions of the maneuverability-agility space: ctenophores' ability to actuate rows at different frequencies allows them to control the sharpness of the turn as well as the forward swimming speed during a turn. This suggests that different strategies may be used for different functional behaviors—for example mode 1 may be used during feeding, when the animal must rotate to maneuver food into its mouth but does not need to swim forward, and mode 2 for escaping or resetting tentacles after feeding—a process that is likely to be primarily driven by fluid drag, and therefore requiring a straight or moderately curved trajectory at a reasonably high speed. We note that in our model, we do not consider the reversibility of the power stroke, which is likely to increase maneuverability even more (for example, if two opposing row-pairs beat with opposite power stroke directions, $\overline{R/L}$ may decrease even further than what the model predicts). We note that due to the structure of the ctenophore statocyst, ctenophores cannot independently control the frequency of all 8 ctene rows, but control the rows in pairs such that each body quadrant (divided by the tentacular and sagittal planes) may carry a unique frequency. Our exploration in Fig 11 therefore shows not what is possible for a behaving animal, but what may be achieved by a bioinspired device or vehicle operating under a similar propulsion system.

This body plan could be used as inspiration for millimeter-scale robotic platforms, with the potential to rapidly reorient into any direction from an initial position. The reduced-order model presented here can be used in the design phase to estimate the general swimming dynamics and inform future robots' control requirements. However, further work should

include the fluid-structure interactions between the appendages/body and the surrounding flows. As high speed videography becomes more accessible, 3D-resolved kinematics may be compared across species (both within the phylum Ctenophora and externally, with other meta-chronal swimmers). Soft robotic models and CFD simulations would provide further tools for a more controllable exploration of metachronal swimming strategies and their concomitant maneuvering capability. However, it is clear that metachronal locomotion–with its scalability, efficiency, and (as we have shown here) high degree of maneuverability and agility—represents a promising new direction for bioinspired technology.

## Supporting information

**S1 Text. Supporting information text.** The supporting text includes details for the solution procedure, the numerical implementation, the formulations for various coefficients, and the validation of the model against experimental data.
(PDF)

**S1 Video. Ctene kinematics vs. reduced-order model approximation.** This video shows a side-to-side comparison of a ctenophore rowing and the reduced-order model approximation.
(MP4)

**S2 Video. Turning mode 1 example.** The video captures turning mode 1, where two adjacent rows beat at a higher frequency than the two opposite rows while the remaining four are inactive.
(MP4)

**S3 Video. Turning mode 2 example.** The video captures turning mode 2, where the four outer rows beat at a higher frequency than the four rows on the opposite side.
(MP4)

**S4 Video. Turning mode 3 example.** The video captures turning mode 3, where six rows beat at a higher frequency than the remaining two.
(MP4)

**S5 Video. Turning mode 4 example.** The video captures turning mode 4, where all rows beat at approximately the same frequency.
(MP4)

**S6 Video. Ctene reversal example.** The video shows a ctenophore switching its swimming direction by reversing the ctenes power stroke direction.
(MP4)

## Acknowledgments

The authors gratefully acknowledge Amy E. Mass, David W. Murphy, Elizabeth K. Seber, Fer-hat Karakas, and Andrea Miccoli for their assistance with data collection at BIOS.

## Author Contributions

**Conceptualization:** Adrian Herrera-Amaya, Margaret L. Byron.

**Data curation:** Adrian Herrera-Amaya.

**Formal analysis:** Adrian Herrera-Amaya, Margaret L. Byron.

**Funding acquisition:** Margaret L. Byron.

**Investigation:** Adrian Herrera-Amaya, Margaret L. Byron.

**Methodology:** Adrian Herrera-Amaya, Margaret L. Byron.

**Project administration:** Margaret L. Byron.

**Software:** Adrian Herrera-Amaya.

**Supervision:** Margaret L. Byron.

**Validation:** Adrian Herrera-Amaya.

**Visualization:** Adrian Herrera-Amaya.

**Writing – original draft:** Adrian Herrera-Amaya, Margaret L. Byron.

**Writing – review & editing:** Adrian Herrera-Amaya, Margaret L. Byron.

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
