## [Decision Letter · Decision Letter 0]

21 Jul 2023

Dear Dr. Byron,

Thank you very much for submitting your manuscript "Metachronal coordination enables omnidirectional swimming *via* spatially distributed propulsion" for consideration at PLOS Computational Biology.

As with all papers reviewed by the journal, your manuscript was reviewed by members of the editorial board and by several independent reviewers. In light of the reviews (below this email), we would like to invite the resubmission of a significantly-revised version that takes into account the reviewers' comments.

We cannot make any decision about publication until we have seen the revised manuscript and your response to the reviewers' comments. Your revised manuscript is also likely to be sent to reviewers for further evaluation.

Sincerely,

Natalia L. Komarova

Section Editor

PLOS Computational Biology

Natalia Komarova

Section Editor

PLOS Computational Biology

Reviewer's Responses to Questions

**Comments to the Authors:**

Reviewer #1: This paper combines unique tools for 3D motion tracking of the cilia-like appendages of comb jellies with a minimalist computational model that seems to capture the dominant fluid-structure interactions during turning. The study is well motivated, and the mathematical methods seem sound and well reasoned (and is a 3D extension of the tested and published 1D work by the authors in Integr Comp Biol 2021). However, I have some questions that need clarification, and some comments/suggestions to make this more suitable for PLoS Comput Bio.

Questions:

(1) A critical piece of information gathered from the experimental snapshots is beat frequency, which then enters the kinematic computational model. How is this data converted to a time-continuous series? I see the fit in supplemental fig.3 but there is no information about how this first is performed, what it captures and what it misses, if there is any filtering etc.

(2) On that note, the data in supplementary fig.3 is unclear: there are two sets of points and two curves. Which curve fits which data? Are the data oscillating between the upper sets and the lower sets of dots (in which case the curves miss the rapid oscillations)? Or are the data separately the upper and lower sets (in which case the fits seem systematically shifted).

(3) There seem to be clusters or groups of blue points In figure 3 of the main text (the MAP diagram). Do these correspond to physically related conditions? One could almost squint and make out 4 straight lines around which most of the points are clustered. Could these correspond to the different modes? If so, could the modality be somehow ‘scaled’ out to collapse the data?

(4) Caption for figure 3 lists (A) and (B) which do not appear in the figure.

(5) While the authors do a good job of clarifying what is not captured in this minimalist model (e.g. hydrodynamic interactions), some of the modeling choices could use more justification. For example, why the choice of the oscillating plate rather than a hinged ‘flapping’ plate which might seem more natural and perhaps simpler to model (i.e, the red dashed line as the plate instead of the green solid line in Fig. 9D)?

(6) A key discussion element is the ‘mode’ of turning. Would it be possible to sketch a cartoon that clearly demarcates the 4 modes that are repeatedly discussed throughout the paper?

(7) A comparison between and discussion around the measured and simulated trajectories (such as in supplementary fig. 4) might be apt in the main text.

Reviewer #2: In this investigation, the authors mainly explored the maneuverable and agile swimming behavior of ctenophores achieved by different metachronal propulsion strategies through 3D high-speed videography and reduced-order mathematical modeling. This topic is interesting and innovative. I have several concerns and recommendations.

1. Some morphometric parameters in Table 1 and Fig. 2 were hard to understand. For example, which rows did the sagittal row and tentacular row refer to, respectively? It is better to give straightforward schematic diagrams to illustrate the sagittal and tentacular plane and all the morphometric parameters for easier read.

2. In the Turning performance part, the authors mentioned that “In the first strategy (mode 1), two adjacent rows … and the two opposite rows … while the remaining four rows are inactive.” However, the “two adjacent rows” and “two opposite rows” have not been identified.

3. In the Omnidirectionality part, the authors stated that “in our observations, animals swam freely (without external stimuli)” and recorded the trajectories of the ctenophore in Fig 4. So, the motion of the ctenophore in space should be random. However, as shown in Fig 4A, the ctenophore mainly moved along the X axis and hardly along the Z axis, which resulted in the elliptical gray cloud. Why did the ctenophore tend to move along the X axis rather than the Z axis?

4. What is the physical meaning of BL/s as the unit of the average speed?

5. The typos should be correct. For example, “via” in line 57 should be written in upright form and “30×30×30 〖mm〗^2” should be “30×30×30 〖mm〗^3”.

Reviewer #3: By combining the 3D imaging system and reduced-order analytical modeling, the authors investigate the swimming performance of ctenophores, and especially focus on the agility evaluated by the swimming speed (unit: body length per second) and the maneuverability evaluated by the average normalized radius of curvature (i.e., turning performance). With the computational simulations and analysis of experiments, the authors conclude that ctenophores can quickly reorient and simultaneously keep a high swimming speed (close to 70% of the observed maximum). Overall, the analytical and experimental methods are scientific, and the results are convincing. However, this paper is not well organized, and some other issues need to be further addressed. Therefore, I suggest accepting this manuscript after a major revision. The comments are illustrated as follows:

1. The title claims that “metachronal coordination enables omnidirectional swimming”. As we know, metachronal coordination, as a cross-scale phenomenon, is ubiquitous [Integr. Comp. Biol. 2021, 61, 1674-1688.]. It has been verified that metachronal motion can extremely enhance the propulsion ability of natural cilia [PNAS 2013, 110, 4470-4475.]. However, the observations in this paper suggest that omnidirectional swimming is mainly due to the combination of (i) different swimming modes (according to actuated rows and adjustable beating frequency to distinguish) and (ii) the flexible forward/reverse swimming of ctenophores. By providing symmetric (mode 4) or asymmetric (mode 1-3) propulsion along the body axis, ctenophores can control themselves to go straight or turn. Even if the propulsive forces are provided from ciliary beating and metachronal coordination, however, the omnidirectional swimming is actually realized by the smart swimming strategies of ctenophores, and is not directly related to metachronal coordination. In particular, when the authors use a reduced-order analytical model that ignores hydrodynamic interactions among propellers, the summary (“metachronal coordination enables omnidirectional swimming”) becomes weird and incomprehensible. Please explain this issue.

2. The authors indicate that 3D high-speed videography is first used to observe ctenophores and past works usually adopt a 2D imaging system. Surely, the discovery of four types of 3D swimming modes is impressive. But does this system have other advantages? It will be great if the author can list a table to compare the experimental conclusions and technical limitations of some previous 2D imaging systems, and which ones are well resolved in the 3D imaging system of this article. And in the “Conclusion” section, an appropriate discussion of this system’s potential for follow-up research is also expected.

3. Again, the authors repeatedly emphasize the importance of the metachrony of ctenophores, please give some related qualitative/quantitative descriptions of their motion parameters, e.g., antiplectic or symplectic, the phase difference of neighboring cilia in the metachronal coordination, etc.

4. Because of the gap between the mathematical model and real biological swimmers, correspondingly, there exists a gap between the computational simulation results and the experimental results. The authors should comment on this, especially when the authors want to throw out the conclusions such as “The model results confirm that ctenophores’ metachronal rowing platform is highly maneuverable and agile, with performance limits that may extend beyond our experimental observations (lines 183-185)”.

5. It is worth noting that when the authors would like to emphasize that the swimming performance is excellent, please make the conclusion more objective and intuitive. For example, lines 167-168 “still a comparatively sharp turn, carried out at 71% of the maximum recorded straight-line swimming speed”. The authors want to use this to indicate that ctenophores keep both high agility and maneuverability. But it seems that the maximum speed is from another individual, and how could the authors rate that 71% is an excellent indicator? Maybe there are some comparisons to other swimmers’ turning movements?

6. I understand the authors want to put some technical details in the “Methods” section for detailed analysis and discussion. But please briefly describe the used methods in the main text, and refer to the specific location in “Methods”. The coherence and logic of various parts in the main text are poor, thus the readability is greatly affected. Please reorganize the manuscript. Also, some tiny problems exist, for example, line 249 directly indicates Figure 7B, but Figure 6 is still not described at this time. The authors do need to make the manuscript clear.

Reviewer #4: This manuscript is nice analysis that uses both experimental work and model simulations to examine how ctenophores can manipulate their ctene rows to maneuver and an examination of their capabilities to turn. It has the potential to reveal a lot about how ctenophore control their maneuverability. However, I think it falls a bit short because the authors could more thoroughly analyze their data and model to better describe the observed capabilities and mechanisms of turning. For example, they mention the different modes of swimming but by quantifying the kinematics more thoroughly they could examine what their observations reveal about how the different modes may be used. The authors speculated about this using the model but they should compare these predictions to observed data. Further the figures should really be revised to better distinguish the different modes plotted (see below for details). Finally, the discussion should be expanded to more fully examine how ctenophores use ctenes to maneuver and want the model reveals how this could be expanded upon for bio-inspired vehicles.

In conclusion, I recommend that the article undergoes major revision before it is acceptable for publication.

Specific comments:

Line 112 spell out Bolinopsis (first time used in Intro)

Line 333 and Blue?

Fig. 3 – Should distinguish the different modes in the figure. Cannot interpret what each blue scatter line represents.

Lines 171-185 – The simulation results description in this paragraph does not add anything to the experimental observations. In experiments Bolinopsis is able to have high maneuverability while being agile. Model just confirms that Bolinopsis is able to do this using its metachronal ctenes. That doesn’t add anything. The ctenes are obviously able to do this.

Fig. 4 – it would be very helpful to place ctenophore icons showing the orientation of bolinopsis at different locations in the MV. Otherwise, very difficult to interpret the MV.

Fig. 5 – display different simulated modes with different colors or line styles.

Model predicts that mode 1 is for high maneuverability while modes 2 and 3 allow for faster agility. You should use kinematic data to confirm these predictions. In other words, you should report the maneuverability and agility of each mode that you observed.

Lines 240-243 – this belongs in the Discussion and should be expanded to discuss what the experimental work and model suggest about how ctenophores a achieve different maneuvering outcomes. This would also be the spot to bring in maximizing R/L by reversing inside ctenes as mentioned in lines 284-286.

Line 256 should be in past tense “we explored”

Line274-277 – This is an interesting observation but needs to be quantified and reported in the Results. What as the mean and max R/L

Lines 284-286 – What is the support for this statement? Literature or your data.

Discussion should be expanded to discuss, based on your experiments and model, how ctene rows can be manipulated to achieve different swimming outcomes (see above). You should also discuss, based on what we know about how ctenophores use their apical organ to control ctene rows, what the what the potential limits are to modes of swimming. Can ctenophores independently control all ctene rows like you simulated in Fig. 6? What can they do?

**Have the authors made all data and (if applicable) computational code underlying the findings in their manuscript fully available?**

Reviewer #1: Yes

Reviewer #2: Yes

Reviewer #3: None

Reviewer #4: Yes

PLOS authors have the option to publish the peer review history of their article (what does this mean?). If published, this will include your full peer review and any attached files.

Reviewer #1: No

Reviewer #2: **Yes: **Zhengzhi Wang

Reviewer #3: No

Reviewer #4: No
---

## [Decision Letter · Decision Letter 1]

11 Oct 2023

Dear Dr. Byron,

We are pleased to inform you that your manuscript 'Omnidirectional propulsion in a metachronal swimmer' has been provisionally accepted for publication in PLOS Computational Biology.

Best regards,

Alison Marsden

Academic Editor

PLOS Computational Biology

Natalia Komarova

Section Editor

PLOS Computational Biology

Reviewer's Responses to Questions

**Comments to the Authors:**

Reviewer #1: The authors have clarified and/or addressed all of my previous comments. I appreciate the attention to detail in their response, and I believe the manuscript is overall much stronger, clearer, and suitable for publication in PLOS Comp Bio.

Reviewer #2: The authors have addressed all of my previous comments.

Reviewer #3: I appreciate the careful revisions of the manuscript. All my concerns/questions are well addressed in full. I recommend the manuscript for publication. Great job!

Reviewer #4: The authoers have done a nice job addressing my review. The ms is now acceptable for publication.

**Have the authors made all data and (if applicable) computational code underlying the findings in their manuscript fully available?**

Reviewer #1: Yes

Reviewer #2: None

Reviewer #3: Yes

Reviewer #4: Yes

PLOS authors have the option to publish the peer review history of their article (what does this mean?). If published, this will include your full peer review and any attached files.

Reviewer #1: No

Reviewer #2: No

Reviewer #3: No

Reviewer #4: No

---

## [Editor Report · Acceptance letter]

9 Nov 2023

PCOMPBIOL-D-23-00110R1 

Omnidirectional propulsion in a metachronal swimmer

Dear Dr Byron,

I am pleased to inform you that your manuscript has been formally accepted for publication in PLOS Computational Biology. Your manuscript is now with our production department and you will be notified of the publication date in due course.

With kind regards,

Zsuzsanna Gémesi
